# Selling one's future: over-indebtedness and the risk of poor mental health and the role of precarious employment – results from the Scania Public Health Cohort, Sweden

Per-Olof Östergren ![ORCID],[1] Theo Bodin,[2,3] Catarina Canivet,[1] Mahnaz Moghaddassi,[1] Andreas Vilhelmsson[4]

**To cite:** Östergren P-O, Bodin T, Canivet C, et al. Selling one's future: over-indebtedness and the risk of poor mental health and the role of precarious employment – results from the Scania Public Health Cohort, Sweden. BMJ Open 2022;**12**:e061797. doi:10.1136/bmjopen-2022-061797

¹Department of Clinical Sciences Malmö, Lund University, Lund, Sweden
²Institute of Environmental Medicine, Karolinska Institutet, Stockholm, Sweden
³Region Stockholm, Center for Occupational and Environmental Medicine, Stockholm, Sweden
⁴Department of Laboratory Medicine, Lund University, Lund, Sweden

**Correspondence to**
Per-Olof Östergren;
per-olof.ostergren@med.lu.se

## ABSTRACT

**Objectives** The credit market has expanded rapidly, increasing the risk of over-indebtedness among those who lack secure employment or adequate income, an issue of concern in the COVID-19 aftermath. We investigated the role of over-indebtedness for developing poor mental health, and whether this impact is modified by age, gender, educational level or being in precarious employment.

**Methods** This is a cohort study using data from the Swedish Scania Public Health Cohort, based on individuals randomly selected from the general adult population in Scania, southern Sweden, initiated in 1999/2000 (response rate 58%) with follow-ups in 2005 and 2010. Over-indebtedness was assessed by combining information on cash margin and difficulty in paying household bills. Mental health was assessed by General Health Questionnaire-12. Those with poor mental health at baseline were excluded, and the analyses were further restricted to vocationally active individuals with complete data on main variables, resulting in 1256 men and 1539 women.

**Results** Over-indebtedness was more common among women, among persons with a low educational level, born abroad and with a precarious employment at baseline. The age-adjusted incidence rate ratio (IRR) for poor mental health in 2010 among individuals exposed to over-indebtedness in 1999/2000 or 2005 was 2.2 (95% CI 1.7 to 2.8). Adjusting for educational level, country of origin and precarious employment in 1999/2000 or 2005, yielded an IRR of 2.0 (95% CI 1.6 to 2.6). An interaction analysis indicated that a high level of education may act synergistically with over-indebtedness, regarding poor mental health among men.

**Conclusions** Over-indebtedness was related to unfavourable societal power relations, regarding social class, gender and foreign birth. Precarious employment was independently linked to poor mental health and may also mediate the effect by over-indebtedness. The COVID-19 pandemic might entail increased over-indebtedness, which should be acknowledged in policies aiming at buffering social effects of the pandemic.

## STRENGTHS AND LIMITATIONS OF THIS STUDY

⇒ This was a prospective cohort study with 2795 vocationally active men and women randomly selected from the general adult population.
⇒ The exposure and outcome variables have been used and validated in previous studies.
⇒ Detailed information concerning employment history and details regarding current employment for each individual was used for determining precarious employment status.
⇒ Non-response and missing data among respondents may have caused some selection bias.
⇒ We lacked access to an objective measure regarding over-indebtedness, which may have caused misclassification.

## INTRODUCTION

Credit plays a paramount role in modern economies, since it increases the possibility to make investments that can generate future financial and other gains and to allocate lifetime income in a more optimal relation to need. From a health perspective, such resources are very likely related to opportunities for improved health over the life-course. However, when the credit market targets those who cannot make ends meet, this can cause increasing indebtedness, which can lead to poorer health, both due to loss of material and other resources and to exposure to growing stress and worries about one's loss of control over the household economy. Thus, the proportion of individuals who are prone to over-indebtedness will most likely expand considerably in the aftermath of the COVID-19 pandemic. Moreover, indebtedness could be expected to have a differential effect on health, much depending on factors such as socioeconomic position, country of birth (ie, migration background) and age.

The credit market has expanded over the last decades leading to a build-up of consumer debt. However, it was not until the financial crisis in 2007–2008 that an increased proportion of households that had difficulties making ends meet was noted in the European Union.[1] As of 2016, the household debt in relation to disposable income was more than twice as high in the OECD compared to 1996.[2] Measured as share of gross domestic product, the indebtedness of Swedish households has doubled since the 1990s.[3] According to public reports, the number of over-indebted individuals in Sweden amounts to between 6% and 18% of the population, depending on definition. The subsequent yearly societal cost is estimated to 30 billion SEK.[4] The credit market has also changed in terms of type of credit providers and types of 'products', for example, so called 'pay-day loans' as a temporary financial solution when financial means are insufficient to cover daily cost of living because of inadequate income and lack of savings.[5] These types of loans have at the same time become very accessible via the Internet and mobile phones. Since the interest rates and fees generally are very high for these loans, this could quickly lead to a spiral of indebtedness where the loan-taker in practice ends up owing the lender a substantial proportion of his/her future income.[6]

Over-indebtedness can be defined as a situation, persisting for a sustained period of time, in which daily cost of living exceeds income, and in which the imbalance cannot be compensated by savings. This will inevitably lead to use of credit in one way or another, the cost of which will add to the discrepancy between expenditure on the one side and income and savings on the other. This situation could represent an emerging significant stressor and determinant of ill health, and thereby constitute a source of health inequality that is likely to be overlooked in research on income or wealth and health, since information concerning over-indebtedness can not easily be retrieved from registers on income, wealth or social transfers.

A growing number of scientific studies on the indirect and direct association between over-indebtedness and health have been published in the scientific literature. Systematic reviews have provided support for the assumption that the financial recession has been associated with an increase in suicide rates in high-income countries such as Europe and North America.[7] Further, socioeconomic mediators such as unemployment, income decline and unmanageable debts are significantly associated with poor mental well-being, increased rates of common mental disorders, substance-related disorders and suicidal behaviours.[8–11] Individuals who cannot make their loan payments have been found to suffer from depression and have suicidal ideation more often compared with those without such problems.[12] Another study investigating predictors of mental health in people on the verge of bankruptcy found that perceived financial strain was an important predictor of poor mental health.[13] However, most of the studies have been cross-sectional, and more longitudinal studies are needed to corroborate the mentioned findings.

High debt relative to available assets has also been associated with poorer self-assessed general health, higher diastolic blood pressure, obesity, poorer health-related behaviour and myocardial infarction.[14] The effect of over-indebtedness on health has been found to be stronger for women than for men.[15] In a life-course perspective, poor self-rated health has been found to be even more prevalent among those who have experienced economic stress both early in life and later on.[16]

According to a systematic review and meta-analysis examining the relationship between personal debt and mental health,[17] ten epidemiological studies with nationally representative samples of the general population were found, whereof seven came from the UK and nine out of ten were cross-sectional. Unsurprisingly, very high ORs for different mental health outcomes were found. In many cases the causality pattern could be described as a vicious circle, which could start either with a decrease in income or in health. Poor mental health provoked by over-indebtedness tends to increase the risk of further financial problems and so forth.[18] Moreover, over-indebtedness or coping with excessive credit is associated with subjective distress such as shame, anxiety and apathy.[19] The mentioned process could involve mediating mechanisms, as well as situations where certain exposures modify the effect of each other, such that already vulnerable groups, for example, individuals with low education or in a precarious employment situation, are hit more severely by over-indebtedness in terms of resulting poor mental health.

To the best of our knowledge, evidence regarding the role of precarious employment as a mediator or modifier of the impact of over-indebtedness on mental health is lacking.

In summary, previous research relies heavily on cross-sectional data from the UK, why studies from other national contexts are needed. Moreover, because of the plausible bidirectional causal pathways, prospective studies of long duration with several assessments of both types of variables, over-indebtedness as well as mental health, are preferable. Also, as mentioned above, little is known about the role of precarious employment and the differential health effects across sociodemographic groups.

## OBJECTIVES

The aim of the study was to investigate the role of over-indebtedness for the development of poor mental health in a longitudinal perspective, and whether this impact is modified by age, gender, educational level, housing tenure or being in a precarious employment situation.

## METHODS
### Study population
The present cohort was established in 1999/2000 and followed up in 2005 and 2010.[20] At baseline, a postal questionnaire was sent out to 25 000 men and women, 18–80 years old. These individuals were randomly selected from the population register, such that equal representation was achieved from all municipalities in the county of Scania (population 1.3 million), Sweden. In total, 13 589 out of 23 437 eligible individuals returned a completed questionnaire (response rate 58%). All of those who responded at baseline and still residing in the county were invited to follow-up after 5 and 10 years. Out of 12 002 respondents alive and still living in the region after 10 years, there were 8206, that is, 68%, who also participated in the 2005 and 2010 inquiries.

A comparison between these 8206 respondents and the corresponding Scania general population at baseline[20] showed that younger individuals, men and individuals with a low educational level were slightly under-represented among respondents. However, a recently published study from this cohort, showed very little proof of selection bias regarding common health outcomes, including mental health.[21]

Since one aim of the study was to look into the relationship between precarious employment and over-indebtedness and mental health, we first excluded individuals who were not vocationally active, that is, at baseline were retired, on disability pension, or on long-term sick leave, and also those who answered that they did not wish to work a year from now (N=1744). Out of the remaining 6462 persons, those with lacking data on employment precariousness (N=2479) were excluded; out of these, 1106 were 55–80 years old at baseline, that is, were qualified for old-age retirement during the studied window of time. Further, we excluded individuals who lacked baseline data regarding the variables included in the multivariable analyses (rather than assigning them as 'internal' missing); that is, housing tenure (N=159) or educational level (N=158), also those lacking baseline and 2005 information on over-indebtedness (N=249), furthermore those who lacked information regarding mental health at all follow-ups (N=161). Finally, in order to reduce the influence of reverse causation, we also excluded individuals with poor mental health at baseline (N=1563), instead of controlling for this in the multivariable analysis. Thus, the final study population consisted of 2795 individuals, 1256 men and 1539 women.

### Outcome variable: poor mental health
Mental health was measured at all three time points of assessment, using the 12-item version of the General Health Questionnaire (GHQ-12). We used the 0–0–1–1 scoring method (range 0–12) recommended by the creators of the instrument,[22] with poor mental health or 'GHQ-caseness' determined by the population mean and defined as a score of 2 or higher.[23]

### Over-indebtedness
Two items have been used previously in several Swedish studies to evaluate economic hardship, namely inability to meet expenses and lacking cash reserves.[24] For the former we used the question 'How often during the last 12 months have you had problems paying your bills?' and for the latter, 'In case of an unforeseen emergency, do you have the capacity to raise 12 000 SEK (approximately 1100 EUR) in a week's time?'. The same questions were asked in 1999/2000, 2005 and 2010, but the specified amount was raised to 14 000 SEK, approximately 1300 EUR, in the 2010 survey. Over-indebtedness was defined as affirming 'about half of the months' (or more often) on the first question and answering 'no' on the second.

### Other variables
Age was classified into four groups, except in the interaction analysis, where it was dichotomised, and in the multivariate analysis, where it was used as a continuous variable. Country of origin was recorded as 'born in Sweden' (yes/no). Educational level at baseline was determined by the self-reported total years of formal education and classified into four groups. Housing tenure was dichotomised as either owned or rented residence. For employment precariousness, we used a dichotomous variable, based on a combination of data on present unemployment, episode of involuntary unemployment during the past 3 years (no/yes), currently temporarily versus permanently employed, and perceived job insecurity.[25 26]

### Statistical methods
The relationships between background factors and poor mental health in 2010 are presented as percentages and age-adjusted IRRs, which are a good estimate of relative risks, using a modified Poisson regression model with robust standard errors.[27] In the multivariate analysis, over-indebtedness was tested against the outcome with the stepwise addition of educational level, country of origin and precarious employment.

The tests for effect modification were performed with simple dummy variables. The synergy indices were calculated as proposed by Rothman.[28] according to which a synergy index >1 may indicate a synergistic effect, and a synergy index <1 an antagonistic effect between two exposure variables. In the interaction analysis, educational level was dichotomised as '12 years or less' versus '13 years or more'. There was a negative relationship between educational level and over-indebtedness (see the Results section), while the relationship with the outcome variable poor mental health at follow-up was positive (see the Results section), and therefore a high educational level was designated as exposure.

Two standard statistical analysis programmes were used, IBM SPSS Statistics for Windows, V.22.0, Armonk, NY: IBM Corp. and Stata Statistical Software: Release V.12, College Station, TX: StataCorp LP.

**Table 1** Background characteristics at baseline in 1999/2000 of 1256 men and 1539 women from the Scania Public Health Cohort, in relation to over-indebtedness

| | | All | | | | Men | | | | Women | | | |
|---|---|---|---|---|---|---|---|---|---|---|---|---|---|
| | | Total | No* | Yes* | % Yes | Total | No | Yes | % Yes | Total | No | Yes | % Yes |
| Total | | 2795 | 2679 | 116 | 4.2 | 1256 | 1221 | 35 | 2.8 | 1539 | 1458 | 81 | 5.3 |
| Age | 18–24 | 207 | 196 | 11 | 5.3 | 96 | 92 | 4 | 4.2 | 111 | 104 | 7 | 6.3 |
| | 25–34 | 621 | 596 | 25 | 4.0 | 278 | 272 | 6 | 2.2 | 343 | 324 | 19 | 5.5 |
| | 35–44 | 750 | 710 | 40 | 5.3 | 324 | 314 | 10 | 3.1 | 426 | 396 | 30 | 7.0 |
| | 45–54 | 966 | 930 | 36 | 3.7 | 430 | 416 | 14 | 3.3 | 536 | 514 | 22 | 4.1 |
| | 55–63 | 251 | 247 | 4 | 1.6 | 128 | 127 | 1 | 0.8 | 123 | 120 | 3 | 2.4 |
| Educational level (years) | 9 or less | 502 | 484 | 18 | 3.6 | 265 | 256 | 9 | 3.4 | 237 | 228 | 9 | 3.8 |
| | 10–12 | 954 | 892 | 62 | 6.5 | 437 | 416 | 21 | 4.8 | 517 | 476 | 41 | 7.9 |
| | 13–14 | 633 | 609 | 24 | 3.8 | 253 | 250 | 3 | 1.2 | 380 | 359 | 21 | 5.5 |
| | 15 or more | 706 | 694 | 12 | 1.7 | 301 | 299 | 2 | 0.7 | 405 | 395 | 10 | 2.5 |
| Born in Sweden | Yes | 2588 | 2494 | 94 | 3.6 | 1163 | 1136 | 27 | 2.3 | 1425 | 1358 | 67 | 4.7 |
| | No | 207 | 185 | 22 | 10.6 | 93 | 85 | 8 | 8.6 | 114 | 100 | 14 | 12.3 |
| Precarious employment | No | 2051 | 2004 | 47 | 2.3 | 943 | 926 | 17 | 1.8 | 1108 | 1078 | 30 | 2.7 |
| | Yes | 744 | 675 | 69 | 9.3 | 313 | 295 | 18 | 5.8 | 431 | 380 | 51 | 11.8 |
| Housing tenure | Owned | 2176 | 2112 | 64 | 2.9 | 990 | 969 | 21 | 2.1 | 1186 | 1143 | 43 | 3.6 |
| | Rented | 619 | 567 | 52 | 8.4 | 266 | 252 | 14 | 5.3 | 353 | 315 | 38 | 10.8 |

*Over-indebted at baseline.

## Patient and public involvement statement

Patient or public involvement in the design, reporting or dissemination of results was not possible.

## RESULTS

Table 1 shows the prevalence of over-indebtedness in the total study population at baseline and also separately by gender. Overall, the prevalence of over-indebted individuals was 4.2%, but with a clear difference between men and women (2.8 vs 5.3%, respectively). The only clear age-related difference in over-indebtedness consisted of a low prevalence in participants aged 55 and above. Over-indebtedness was more common in those with an educational level of 10–12 years, versus those with less or with more years. When educational level was dichotomised as '12 years or less' versus '13 years or more', 5.5% of those with a low educational level were over-indebted at baseline, versus 2.7% of those with a high educational level (data not shown in tables). Over-indebtedness was three times as common among those born in another country than Sweden. As perhaps expected, over-indebtedness was four times as prevalent among those who reported a precarious employment situation versus those with a non-precarious situation and almost three times as prevalent among those who rented their dwelling (compared with those who owned it). The gender pattern was largely similar for all factors.

Table 2 shows the results of the longitudinal analysis, where poor mental health at follow-up in 2010 was used as the outcome. The age-adjusted risk for poor mental health at follow-up in over-indebted individuals, measured as IRR, was 2.2 (95% CI 1.7 to 2.8) in the entire group, 2.6 (1.8 to 3.9) for men and 1.9 (1.4 to 2.6) for women. The risk for poor mental health was four times higher in those who were 18–24 years at baseline, with an IRR of 4.1 (2.3 to 7.4), and decreased successively with increasing age, compared with those who were 55 or older at baseline, which was used as the reference group. That pattern was similar in men and women. Those with an educational level of 13–14 years at baseline had a slightly higher risk of poor mental health at follow-up (IRR=1.4; 95% CI 1.01 to 2.0) than those with 9 years or less, which was used as the reference. When educational level was dichotomised, 15.1% of those with a high educational level had poor mental health at follow-up versus 13.9% of those with a low educational level (data not shown in tables). Foreign birth was related to poor mental health in men only, with an IRR of 1.6 (1.01 to 2.6). Having a precarious employment situation during the period 1999/2000 to 2005 was statistically significantly related to poor mental health in 2010 (IRR=1.4; 95% CI 1.2 to 1.7), although also here with a gender difference; the IRR for men was 1.8 (1.3 to 2.4) and for women 1.2 (0.95 to 1.5). Housing tenure (rented dwelling) at baseline was not related to poor mental health in 2010.

Table 3 shows the effect on the risk estimate for poor mental health at follow-up when exposed to over-indebtedness during the period 1999/2000 to 2005, with a stepwise introduction of the main covariates. In the final step, when precarious employment situation during

Östergren P-O, *et al. BMJ Open* 2022;**12**:e061797. doi:10.1136/bmjopen-2022-061797

**Table 2** Background characteristics from 1999/2000 through 2005 in relation to poor mental health in 2010

| | All | | | | Men | | | | Women | | | |
|---|---|---|---|---|---|---|---|---|---|---|---|---|
| | N cases poor mental health* | % cases poor mental health | IRR | 95% CI | N cases poor mental health | % cases poor mental health | IRR | 95% CI | N cases poor mental health | % cases poor mental health | IRR | 95% CI |
| Total | 404 | 14.5 | | | 144 | 11.5 | | | 260 | 16.9 | | |
| Over-indebtedness in 1999/2000 or 2005, at least once — No | 348 | 13.3 | 1 | | 124 | 10.4 | 1 | | 224 | 15.7 | 1 | |
| Yes | 56 | 31.8 | 2.2 | 1.7 to 2.8 | 20 | 31.3 | 2.6 | 1.8 to 3.9 | 36 | 32.1 | 1.9 | 1.4 to 2.6 |
| Age in 1999/2000 — 18–24 | 44 | 21.3 | 4.1 | 2.3 to 7.4 | 17 | 17.7 | 5.7 | 2.0 to 16.3 | 27 | 24.3 | 3.3 | 1.6 to 6.8 |
| 25–34 | 125 | 20.1 | 3.9 | 2.2 to 6.8 | 46 | 16.5 | 5.3 | 2.0 to 14.4 | 79 | 23.0 | 3.1 | 1.6 to 6.1 |
| 35–44 | 114 | 15.2 | 2.9 | 1.7 to 5.1 | 41 | 12.7 | 4.0 | 1.5 to 11.1 | 73 | 17.1 | 2.3 | 1.2 to 4.5 |
| 45–54 | 108 | 11.2 | 2.2 | 1.2 to 3.8 | 36 | 8.4 | 2.7 | 0.97 to 7.4 | 72 | 13.4 | 1.8 | 0.9 to 3.6 |
| 55–63 | 13 | 5.2 | 1 | | 4 | 3.1 | 1 | | 9 | 7.3 | 1 | |
| Educational level in 1999/2000 (years) — 9 or less | 47 | 9.4 | 1 | | 18 | 6.8 | 1 | | 29 | 12.2 | 1 | |
| 10–12 | 155 | 16.2 | 1.3 | 0.9 to 1.8 | 61 | 14.0 | 1.4 | 0.8 to 2.5 | 94 | 18.2 | 1.1 | 0.8 to 1.7 |
| 13–14 | 107 | 16.9 | 1.4 | 1.01 to 2.0 | 27 | 10.7 | 1.2 | 0.7 to 2.1 | 80 | 21.1 | 1.4 | 0.9 to 2.1 |
| 15 or more | 95 | 13.5 | 1.2 | 0.8 to 1.6 | 38 | 12.6 | 1.4 | 0.8 to 2.5 | 57 | 14.1 | 0.9 | 0.6 to 1.4 |
| Born in Sweden — Yes | 369 | 14.3 | 1 | | 128 | 11.0 | 1 | | 241 | 16.9 | 1 | |
| No | 35 | 16.9 | 1.2 | 0.9 to 1.6 | 16 | 17.2 | 1.6 | 1.01 to 2.6 | 19 | 16.7 | 1.0 | 0.6 to 1.5 |
| Precarious employment in 1999/2000 or 2005, at least once — No | 211 | 12.0 | 1 | | 69 | 8.5 | 1 | | 142 | 15.0 | 1 | |
| Yes | 193 | 18.6 | 1.4 | 1.2 to 1.7 | 75 | 16.7 | 1.8 | 1.3 to 2.4 | 118 | 20.0 | 1.2 | 0.95 to 1.5 |
| Housing tenure in 1999/2000 — Owned | 292 | 13.4 | 1 | | 101 | 10.2 | 1 | | 191 | 16.1 | 1 | |
| Rented | 112 | 18.1 | 1.1 | 0.9 to 1.4 | 43 | 16.2 | 1.3 | 0.9 to 1.9 | 69 | 19.5 | 1.0 | 0.8 to 1.3 |

*Poor mental health is defined as GHQ-caseness.
IRR, incidence rate ratio, age-adjusted.

**Table 3** Over-indebtedness at least once in 1999/2000 or 2005 in relation to the outcome poor mental health at follow-up in 2010, with forward stepwise addition of potential confounders; educational level in 1999/2000, country of origin, and precarious employment at least once in 1999/2000 or 2005

| | | Model 1 Age-adjusted | | Model 2 Model 1+adjusted for educational level in 1999/2000 | | Model 3 Model 2+adjusted for country of origin | | Model 4 Model 3+adjusted for precarious employment in 1999/2000 or 2005 | |
| --- | --- | --- | --- | --- | --- | --- | --- | --- | --- |
| | | IRR | 95% CI | IRR | 95% CI | IRR | 95% CI | IRR | 95% CI |
| All | Over-indebtedness | Yes vs no | 2.2 | 1.7 to 2.8 | 2.2 | 1.7 to 2.8 | 2.2 | 1.7 to 2.8 | 2.0 | 1.6 to 2.6 |
| | Educational level | 9 years or less | | | 1 | | 1 | | 1 | |
| | | 10–12 years | | | 1.2 | 0.9 to 1.7 | 1.2 | 0.9 to 1.7 | 1.3 | 0.9 to 1.8 |
| | | 13–14 years | | | 1.4 | 1.03 to 2.0 | 1.4 | 1.03 to 2.0 | 1.5 | 1.1 to 2.1 |
| | | 15 years or more | | | 1.2 | 0.9 to 1.7 | 1.2 | 0.9 to 1.7 | 1.3 | 0.9 to 1.8 |
| | Born in Sweden | No vs yes | | | | | 1.1 | 0.8 to 1.4 | 1.0 | 0.7 to 1.4 |
| | Precarious employment | Yes vs no | | | | | | | 1.3 | 1.1 to 1.6 |
| Men | Over-indebtedness | Yes vs no | 2.6 | 1.8 to 3.9 | 2.7 | 1.8 to 4.2 | 2.6 | 1.7 to 4.0 | 2.2 | 1.4 to 3.5 |
| | Educational level | 9 years or less | | | 1 | | 1 | | 1 | |
| | | 10–12 years | | | 1.4 | 0.8 to 2.4 | 1.4 | 0.8 to 2.4 | 1.4 | 0.8 to 2.4 |
| | | 13–14 years | | | 1.2 | 0.7 to 2.2 | 1.3 | 0.7 to 2.2 | 1.2 | 0.7 to 2.2 |
| | | 15 years or more | | | 1.6 | 0.9 to 2.8 | 1.6 | 0.9 to 2.7 | 1.6 | 0.9 to 2.8 |
| | Born in Sweden | No vs yes | | | | | 1.3 | 0.8 to 2.1 | 1.2 | 0.7 to 1.9 |
| | Precarious employment | Yes vs no | | | | | | | 1.6 | 1.2 to 2.3 |
| Women | Over-indebtedness | Yes vs no | 1.9 | 1.4 to 2.6 | 1.9 | 1.4 to 2.5 | 1.9 | 1.4 to 2.6 | 1.8 | 1.4 to 2.5 |
| | Educational level | 9 years or less | | | 1 | | 1 | | 1 | |
| | | 10–12 years | | | 1.1 | 0.7 to 1.7 | 1.1 | 0.7 to 1.7 | 1.1 | 0.8 to 1.7 |
| | | 13–14 years | | | 1.4 | 0.9 to 2.1 | 1.4 | 0.9 to 2.1 | 1.4 | 0.9 to 2.1 |
| | | 15 years or more | | | 1.0 | 0.6 to 1.5 | 1.0 | 0.6 to 1.5 | 1.0 | 0.6 to 1.5 |
| | Born in Sweden | No vs yes | | | | | 0.9 | 0.6 to 1.4 | 0.9 | 0.6 to 1.3 |
| | Precarious employment | Yes vs no | | | | | | | 1.1 | 0.9 to 1.4 |

IRR, incidence rate ratio.

Östergren P-O, *et al. BMJ Open* 2022;**12**:e061797. doi:10.1136/bmjopen-2022-061797

1999/2000 to 2005 was introduced into the analysis, the risk estimate for poor mental health regarding exposure to over-indebtedness 1999/2000 to 2005 decreased about 10% for the entire group. The decrease in IRR in the fully adjusted model was slightly larger for men than for women. As a general remark, we find it possible that the decrease in IRR is a case of over-adjustment, since all the introduced covariates could be included in the same causal chains, as well as in separate ones, and we have no way to disentangle this.

As it is of considerable interest to investigate whether there are certain segments of the population that are more vulnerable concerning the risk of developing poor mental health when exposed to over-indebtedness, we examined this concerning age and level of education. As seen in table 4A, there was no indication of a synergistic effect of young age and over-indebtedness on the risk of poor mental health at follow-up. However, as seen in table 4B, while there was likewise no interaction between a high educational level and over-indebtedness in women, a higher risk for poor mental health was seen in men with this particular combination of risk factors. The IRR for men was 3.8 (95% CI 1.8 to 8.0), and the resulting synergy index 1.8.

## DISCUSSION

This study confirmed that vocationally active individuals who are exposed to over-indebtedness have an increased risk of developing poor mental health. Age-adjusted estimates showed that the risk was almost tripled for men and doubled for women in this general adult population sample of vocationally active individuals (18–65 years). Considering that about 4% of all individuals in the cohort were exposed to over-indebtedness at baseline and probably substantially more during the follow-up period, over-indebtedness should be regarded as an important general population risk factor for poor mental health. This is pertinent especially in the aftermath of the COVID-19 pandemic, during which the number of vulnerable individuals in the adult population is likely to increase considerably.

At the same time, especially from a health equity perspective, it must be acknowledged that the prevalence and to some extent the impact of exposure to over-indebtedness in the population depend on sociodemographic circumstances. Previous research provides few examples of how this could be integrated into a coherent theoretical framework regarding health equity. We speculate that the reason that younger individuals are more exposed to over-indebtedness than middle-aged or older ones may partly have a 'life-course' explanation (mortgage loans are taken early in life and paid off over time), but there is also a generational effect due to the expansion of credit markets over time. For the other group disparities in over-indebtedness, reasons must be sought elsewhere. Women were more exposed than men, foreign-born more than those born in Sweden, and those with a low

**Table 4A** Interaction analyses, by gender, with synergy indices regarding age group in 1999/2000 and over-indebtedness at least once in 1999/2000 or 2005, in relation to poor mental health in 2010

| Age group in 1999/2000 (18–34 vs 35–63) and over-indebtedness in 1999/2000 or 2005 | All | | | | | Men | | | | | Women | | | | |
|---|---|---|---|---|---|---|---|---|---|---|---|---|---|---|---|
| | N | % cases poor mental health | IRR | 95% CI | SI | N | % cases poor mental health | IRR | 95% CI | SI | N | % cases poor mental health | IRR | 95% CI | SI |
| Older and no over-indebtedness | 1859 | 11.0 | 1 | | | 844 | 8.4 | 1 | | | 1015 | 13.2 | 1 | | |
| Older and over-indebtedness | 108 | 27.8 | 2.5 | 1.8 to 3.5 | | 38 | 26.3 | 3.1 | 1.8 to 5.6 | | 70 | 28.6 | 2.2 | 1.5 to 3.2 | |
| Younger and no over-indebtedness | 760 | 18.8 | 1.7 | 1.4 to 2.1 | | 348 | 15.2 | 1.8 | 1.3 to 2.5 | | 412 | 21.8 | 1.7 | 1.3 to 2.1 | |
| Younger and over-indebtedness | 68 | 38.2 | 3.5 | 2.5 to 4.8 | 1.1 | 26 | 38.5 | 4.6 | 2.7 to 7.8 | 1.2 | 42 | 38.1 | 2.9 | 1.9 to 4.4 | 1.0 |
| | 2795 | | | | | 1256 | | | | | 1539 | | | | |

IRR, incidence rate ratio; SI, synergy index.

**Table 4B** Interaction analyses, by gender, with synergy indices regarding educational level in 1999/2000 and over-indebtedness at least once in 1999/2000 or 2005, in relation to poor mental health in 2010

| Educational level in 1999/2000 and over-indebtedness in 1999/2000 or 2005 | All | | | | | Men | | | | | Women | | | | |
|---|---|---|---|---|---|---|---|---|---|---|---|---|---|---|---|
| | N | % cases poor mental health | IRR | 95% CI | SI | N | % cases poor mental health | IRR | 95% CI | SI | N | % cases poor mental health | IRR | 95% CI | SI |
| Low and no over-indebtedness | 1336 | 12.4 | 1 | | | 649 | 9.9 | 1 | | | 687 | 14.8 | 1 | | |
| Low and over-indebtedness | 120 | 30.0 | 2.2 | 1.6 to 3.0 | | 53 | 28.3 | 2.5 | 1.5 to 4.1 | | 67 | 31.3 | 2.0 | 1.3 to 2.9 | |
| High and no over-indebtedness | 1283 | 14.2 | 1.1 | 0.9 to 1.3 | | 543 | 11.0 | 1.1 | 0.8 to 1.5 | | 740 | 16.5 | 1.1 | 0.9 to 1.4 | |
| High and over-indebtedness | 56 | 35.7 | 2.6 | 1.8 to 3.8 | 1.2 | 11 | 45.5 | 3.8 | 1.8 to 8.0 | 1.8 | 45 | 33.3 | 2.1 | 1.3 to 3.2 | 1.0 |
| | 2795 | | | | | 1256 | | | | | 1539 | | | | |

IRR, age-adjusted incidence rate ratio; SI, synergy index.

educational level more than those with a high level. In short, it seems that the risk for over-indebtedness follows all the main societal power structures, that is, gender, social class and ethnicity. Therefore, it is an important factor to consider when formulating and implementing policies for promoting health equity.

Regarding socioeconomic factors that logically should be strongly related to over-indebtedness, that is, precarious employment and housing ownership (ie, non-ownership), we found that both these factors were strongly related to over-indebtedness, but that only precarious employment was also related to the risk of developing poor mental health during our observation period. However, when we performed an analysis to control for confounding from precarious employment, the impact of over-indebtedness on poor mental health was only moderately reduced. Considering a bidirectional relation between over-indebtedness and poor mental health, the choice to exclude individuals with poor mental health at baseline could have yielded an underestimation of the true association between the variables.

Health inequality could be the result of both differential exposure and differential impact when once exposed.[29] From the discussion above, it seems obvious that there is a clear case of differential exposure regarding over-indebtedness and sociodemographic factors. When we tested by means of two-way interactions whether we in addition could find any evidence for a differential impact, we concluded that this was not the case regarding age (table 4A). We noted, however, that the combination of a high educational level and over-indebtedness (table 4B) was associated with a synergy index of 1.0 in women, whereas the corresponding index in men was 1.8, thus indicating that these factors may act synergistically in a detrimental way for men. One could speculate that over-indebtedness may be especially hard to endure for a well-educated man, since over-indebtedness implies that one is financially not 'in control', whereas gender-based stereotypes of masculinity imply the opposite. Added to that might be the frustration caused by having invested many years in education and not being 'rewarded' by economic stability. Nevertheless, this finding relies on an analysis in which the number of exposed men is small, why the CIs are large. Therefore, this result needs to be replicated in other research.

As shown in many studies as well as in this one, the basic power relations in society, among which would be those determined by gender, social class and ethnicity, show a strong relation to the risk of becoming over-indebted. This is not surprising, because all the above mentioned power relations determine access to factors that could avoid a short-term or long-term situation where credit is needed for securing stable living conditions. Not being able to do so will induce both a loss of control and a loss of material resources, situations that have been shown to increase individuals' susceptibility to poor health development, especially poor mental health. However, the 'Faustian deal' of using credit in this situation could make

Östergren P-O, *et al. BMJ Open* 2022;**12**:e061797. doi:10.1136/bmjopen-2022-061797

things worse, since a spiral of failing credit and declining mental health could place an individual in a situation where he or she in fact has sold a considerable part of future earnings to a moneylender institute. The possibility of having such debts cancelled in a regulated way, which is open to business enterprises through the legal bankruptcy framework, is very restricted for individuals in the Swedish context. This is particularly problematic in a health equity perspective because of the strong association between societal power structures and the risk of becoming over-indebted, all of which could be magnified in the wake of the COVID-19 pandemic.

The results of this study are therefore very relevant for policy, not only regarding public health and health equity, but also for policy formulation and implementation in the judicial and social realms. A specific policy recommendation is to expand the possibilities to write off debt for the individual, in a similar way that now is possible for business enterprises, in order to prevent a vicious circle of poor mental health and increasing debt, which seems maleficent for all involved stakeholders. Because of the growing problem with over-indebtedness, it seems urgent to develop the information systems concerning individual's and household's financial situation, in a manner that high quality information becomes available, not only concerning income but equally concerning financial assets and debt. This, especially since increasing inequality appears to be better mirrored by the latter, rather than the former.

## Strengths and weaknesses

This study uses a prospective design using a large randomly selected general population sample of individuals in the occupationally active age-bracket in southern Sweden. A first selection took place in the very establishment of the Scania Public Health Cohort, since the response rate was 58%.[30] This opens up for the introduction of selection bias. However, two studies that were specifically designed to elucidate selection bias in this cohort concluded that apart from an under-representation of foreign-born individuals, the responders were not statistically different from non-responders regarding other sociodemographic characteristics,[20] nor were any signs of selection bias evident since the risk estimates for common illnesses (including mental health) were very similar between participants and drop outs.[21 31] In the former study, it was also noted that healthcare usage did not differ between responders and non-responders during the baseline survey year. Moreover, such bias could in fact give rise to an underestimation, rather than an overestimation, of the observed differences, since it seems reasonable that heavily indebted individuals for several reasons (lack of postal address, etc) could be under-represented in the sample, especially if their mental health was poor.

The main exposure, over-indebtedness, was based on two questions that in combination have been deemed valid by other authors[24] to identify individuals with a financial situation indicating that their income for a longer period of time was insufficient to meet their regular household bills without having other financial assets (the definition of over-indebtedness). We believe that this measure might be more objective, since it builds on two factual circumstances, than measures solely based on a subjective assessment of financial stress. The prevalence of estimated over-indebtedness in our sample seems to be reasonably on par with the prevalence of over-indebtedness in the general Swedish population at the time of assessment.[32] A direct measure of over-indebtedness would be preferable, but this seems difficult to implement either by register studies or by surveys, since there is a whole range of credit forms to be covered, from regular bank loans to personal or even criminal types of credit that are very difficult to assess in a reliable and valid manner.

Our measure of precarious employment is based on relevant indicators and has been used in a recent study[25]; however, it has not been validated in relation to other existing instruments. Given that confounding from a precarious employment situation was rather weak, and considering that this factor very well could be involved in the same causal chain as the main exposure, we find it unlikely that the discussed possible misclassification had biased our finding to any degree of importance. Although we have assessed potential confounding from main socioeconomic background factors known to be linked to over-indebtedness, such as gender, educational level, age and migration status, we cannot fully rule out residual confounding from other factors.

This study was based on self-reported data. However, the GHQ instrument has been extensively validated, and recent studies carried out in Sweden have shown that the GHQ-12 performs acceptably in detecting depression in the general population[33] and also performs acceptably in discriminating out-patients with depression and anxiety from healthy controls.[34]

## Conclusions

This prospective study showed that exposure to over-indebtedness had a negative effect on vocationally active individual's mental health. It also showed that over-indebtedness is not uncommon in the general working population, and it therefore seems to be a significant contributor to poor mental health at the population level, especially since loss of employment and secure income is likely to increase considerably in the aftermath of COVID-19. Moreover, over-indebtedness was found to be clearly related to an unfavourable position with regard to societal power relations. The findings are therefore of importance not only for policy makers in the area of public health and health equity, but also for policy areas that determine the risk for becoming over-indebted, such as the social and judicial realms.

**Acknowledgements** Preliminary results from this study were presented by P-OÖ at the 12th European Public Health Conference in Stockholm, 1–4 November 2017. The authors would like to thank Elizabeth Cantor-Graae for language consulting.

## Open access

**Contributors** The study was conceived and designed by P-OÖ, CC and AV. The analysis was done by MM. TB drafted the first version of the manuscript. All authors took part in interpreting the results and writing subsequent drafts. P-OÖ is the guarantor, i.e. the author responsible for the overall content of this work.

**Funding** This work was funded by FORTE (Swedish Research Council for Health, Working Life and Welfare; Grant Number 2013-1269) and the Pufendorf Institute at Lund University (Credit Society Project). The funding institutions had no role in the design of the study, data collection, analysis and interpretation of data, or in writing of the manuscript.

**Competing interests** None declared.

**Patient and public involvement** Patients and/or the public were not involved in the design, or conduct, or reporting, or dissemination plans of this research.

**Patient consent for publication** Not applicable.

**Ethics approval** This study involves human participants and was approved by Ethics name ID:The Regional Ethical Review Board at Lund University, Sweden, ID numbers: 1999-99; 2005-471 and 2010-392. Participants gave informed consent to participate in the study before taking part.

**Provenance and peer review** Not commissioned; externally peer reviewed.

**Data availability statement** Data are available upon reasonable request.

**ORCID iD**
Per-Olof Östergren http://orcid.org/0000-0002-7903-6668

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
