## [Reviewer comments · BMJ Open]

ARTICLE DETAILS

TITLE (PROVISIONAL)	Selling one's future: Over-indebtedness and the risk of poor mental health and the role of precarious employment – results from the Scania Public Health Cohort, Sweden
AUTHORS	Östergren, Per-Olof; Bodin, Theo; Canivet, Catarina; Moghaddassi, Mahnaz; Vilhelmsson, Andreas

VERSION 1 – REVIEW

REVIEWER	Thomson, Rachel University of Glasgow
REVIEW RETURNED	22-Mar-2022

GENERAL COMMENTS	This is an interesting study using longitudinal cohort data to examine the effect of over-indebtedness on mental health outcomes in a Swedish population. The analysis appears to be well done, and the findings are novel and of interest. However, I do have some comments and concerns which largely relate to the presentation of the results, discussion of limitations, and a need to put the findings more clearly in the context of existing literature. MAJOR COMMENTS 1) I think the Introduction could be tightened up before publication. A few specific points I would suggest: a) Can you define early in the paper what exactly is meant by over-indebtedness? You mention that there are several definitions, but it might be useful to expand slightly on what these are to provide context for unfamiliar readers. b) In general I felt the number of references seemed low, and the lack of any citations in the first paragraph stands out as particularly unusual – can you add references for where your ideas come from here? E.g., you mention that indebtedness may have differential effects across groups, but don't cite evidence to support this hypothesis. c) You may wish to include some discussion of the similarities and differences between your exposure and related others which have been considered in the literature you are discussing (e.g. perceived financial strain, economic stress, income decline) to make clear your hypotheses on how/why these factors may exert similar (or differing effects) on mental health. d) I'm not sure you can make assertions about the likely 'causality pattern' of the relationship between over-indebtedness and mental health based on the systematic review you cite, given that 90% of the included studies were cross-sectional. It might be more straightforward (and better make the case for your own research) to state that the possibility of reverse causation cannot be ruled
--

	out without longitudinal data, and previous work has not sufficiently done this. e) Clearer justification for your selection of potential effect modifiers (drawing on existing literature) would provide useful context for the reader. 2) Your final sample size is considerably smaller than the initial random sample (~10%), and far less likely to be representative, particularly with the exclusion of those with pre-existing mental health problems and those missing data on your variables of interest. While you do discuss this in the limitations section, it does not necessarily follow that just because studies have found no clear evidence of selection bias in the sample who initially responded to the cohort invitation (58% of the total), the same would be true for your additionally restricted sample. I would therefore either tone down or remove the sentence “It is therefore our assumption that selection bias did not play a major role in the findings currently obtained” and be clearer throughout the paper about the potential implications of selection bias and missing data for the external validity of your findings. It may also be worth referencing the size of the initial sample or the percentage missingness in the Abstract to signpost this to the reader, as currently this reads as though your analytic sample was still representative/random 3) I was surprised that you used a GHQ-12 threshold of two points to indicate poor mental health or ‘caseness’ – I have more often seen a score of four used in the literature, with three sometimes used as a lower cut-off for sensitivity analysis. It would be helpful to more explicitly justify this choice in the Methods section e.g., if this is the more commonly used cut-off in the Swedish population, or if you based your choice on the mean GHQ score in your sample. If possible, I would also suggest running a sensitivity analysis using a different cut-off to provide confidence that your results are not being unduly influenced by the use of the low threshold. 4) In your Results and Discussion, you should be more cautious in interpreting the stratified IRRs where the confidence intervals have considerable overlap (e.g., for age), and acknowledge that this lack of precision creates difficulties for interpretation of the stratified estimates in your Limitations. This also applies to the IRRs in Tables 4A and 4B, particularly since you do not have confidence intervals around the synergy indices and the confidence intervals again largely overlap (they are especially wide around the estimate for high education/over-indebtedness in men, which should be more explicitly acknowledged when drawing inference from this finding in the Discussion). 5) Aside from selection bias, I think the potential for residual confounding is the biggest risk to causal interpretation of your findings, but this is not allocated much discussion in the paper (in fact you consider that instead you may be *over*-adjusting). Based on the literature on income/wealth and health, I suspect there are differences between those exposed and unexposed to over-indebtedness which are not wholly captured by your measured variables of gender, education, country of origin and precarious employment (household composition, individual/household income, area-level deprivation, benefit claimant status,
--	--

	personality/values etc.). More explicit acknowledgement of this as a potential limitation would be welcome. 6) As with the Introduction, there should be more engagement with existing literature in the Discussion, putting your findings into the context of existing research rather than simply restating them. You have some very interesting theories about what your results mean, but I wanted to know more about how these fit with what is already known or theorised e.g., has your 'life-course' explanation for over-indebtedness patterning by age been reported on elsewhere? What is already known about how over-indebtedness is patterned by social power structures? Are there other literatures on how economic exposures may impact the health of well educated people (particularly men) more severely than those with less education? MINOR COMMENTS 1) On the whole, I think the paper should be proofread a little more finely prior to resubmission as I did notice a few minor errors throughout, for example: a) In the first line there is a typo – I think economy should be economies? b) In the opening paragraph, the wording should read 'from' a health perspective rather than 'in' c) 'Data' should be plural throughout d) Covid-19 is sometimes capitalised and sometimes not 2) In the Abstract: a) You should make clear that your sample only includes individuals who are vocationally active, rather than the whole general adult population. b) You could make clearer the interpretation of the final sentence in Results: in what direction does the synergistic relationship between high levels of education and over-indebtedness impact men? c) You mention in the Conclusions section that you found precarious employment to be a potential mediator, but you don't refer to this finding in your included Results? 3) You may wish to include the data not shown in tables (e.g. results dichotomised by education level) in an online supplement, if permitted by the journal. 4) In the second-last paragraph of the Results, I would tone down the strength of your statement on over-adjustment after adding the job insecurity variable from being 'very likely' to 'possible' or something else less strong, as I'm not altogether convinced that the bias couldn't theoretically go in either direction
--	---

REVIEWER	Quinn, Amity University of Calgary
REVIEW RETURNED	07-May-2022

GENERAL COMMENTS	Thank you for the opportunity to review this prospective cohort study on the association of over-indebtedness and mental health. The study is interesting and well-done. I have a few minor comments:
---

	In the strengths and limitations section, please include which general adult population (e.g., southern Sweden). Methods  - At which assessments (time points) were over-debtedness defined? - Statistical analysis: Could you add some additional details about the interaction analyses? Discussion:  - Because information on over-indebtedness is not easily retrievable from registry or administrative health or financial data as you point out, could you expand on what types of policy interventions might be possible to address these problems? - Could you provide some additional discussion on the generalizability of your findings and the policy context as well as policy implications outside of Sweden? Relatedly, I am curious about safety net policies in Sweden, including inside and outside of the health care system.
--	--

VERSION 1 – AUTHOR RESPONSE

Reviewer: 1

Comments to the Author:

This is an interesting study using longitudinal cohort data to examine the effect of over-indebtedness on mental health outcomes in a Swedish population. The analysis appears to be well done, and the findings are novel and of interest. However, I do have some comments and concerns which largely relate to the presentation of the results, discussion of limitations, and a need to put the findings more clearly in the context of existing literature.

MAJOR COMMENTS

1) I think the Introduction could be tightened up before publication. A few specific points I would suggest:

- a) Can you define early in the paper what exactly is meant by over-indebtedness? You mention that there are several definitions, but it might be useful to expand slightly on what these are to provide context for unfamiliar readers.

We are very grateful for this comment, which made us reflect once more over our phrasing regarding the definition of the concept of over-indebtedness. The following sentence has been added in the Introduction section on page 5.

(All page numbers refer to the "Marked-copy" version of the manuscript.)

Over-indebtedness can be defined as a situation, persisting for a sustained period of time, in which daily cost of living exceeds income, and in which the imbalance cannot be compensated by savings. This will inevitably lead to use of credit in one way or another, the cost of which will add to the discrepancy between expenditure on the one side and income and savings on the other.

- b) In general I felt the number of references seemed low, and the lack of any citations in the first paragraph stands out as particularly unusual – can you add references for where your ideas come from here? E.g., you mention that indebtedness may have differential effects across groups, but don't cite evidence to support this hypothesis.

The first paragraph is merely an attempt based on logical reasoning, to frame the potential importance of over-indebtedness as one of the mechanisms behind growing economic inequalities, and the potential adverse health effects of this. It is intentionally therefore not stated as a matter of fact. To our knowledge, there are very few empirical studies addressing inequality in this context.

- c) You may wish to include some discussion of the similarities and differences between your exposure and related others which have been considered in the literature you are discussing (e.g. perceived financial strain, economic stress, income decline) to make clear your hypotheses on how/why these factors may exert similar (or differing effects) on mental health.

This is of course very interesting, however the format of the manuscript does essentially not allow too much discussion outside the methods actually used in the analysis. However, a sentence concerning this has been included in the Discussion section on page 22.

We believe that this measure might be more objective, since it builds on two factual circumstances, than measures solely based on a subjective assessment of financial stress.

- d) I'm not sure you can make assertions about the likely 'causality pattern' of the relationship between over-indebtedness and mental health based on the systematic review you cite, given that 90% of the included studies were cross-sectional. It might be more straightforward (and better make the case for your own research) to state that the possibility of reverse causation cannot be ruled out without longitudinal data, and previous work has not sufficiently done this.

We are very grateful for this comment, and the suggested change has been made and a sentence has been included at the end of the mentioned paragraph on page 5

However, most of the studies have been cross-sectional, and more longitudinal studies are needed to corroborate the mentioned findings.

- e) Clearer justification for your selection of potential effect modifiers (drawing on existing literature) would provide useful context for the reader.

We are grateful for this comment. The following sentence has been added in the Introduction section on page 6.

The mentioned process could involve mediating mechanisms, as well as situations where certain exposures modify the effect of each other, such that already vulnerable groups, e.g. individuals with low education or in a precarious employment situation, are hit more severely by over-indebtedness in terms of resulting poor mental health.

2) Your final sample size is considerably smaller than the initial random sample (~10%), and far less likely to be representative, particularly with the exclusion of those with pre-existing mental health problems and those missing data on your variables of interest. While you do discuss this in the limitations section, it does not necessarily follow that just because studies have found no clear evidence of selection bias in the sample who initially responded to the cohort invitation (58% of the total), the same would be true for your additionally restricted sample. I would therefore either tone down or remove the sentence "It is therefore our assumption that selection bias did not play a major role in the findings currently obtained" and be clearer throughout the paper about the potential implications of selection bias and missing data for the external validity of your findings. It may also be worth referencing the size of the initial sample or the percentage missingness in the Abstract to signpost this to the reader, as currently this reads as though your analytic sample was still representative/random

We thank the reviewer for this relevant comment and the mentioned sentence has been removed from the Strengths and Weaknesses section on page 22 and the suggested information has been added in the abstract.

3) I was surprised that you used a GHQ-12 threshold of two points to indicate poor mental health or 'caseness' – I have more often seen a score of four used in the literature, with three sometimes used as a lower cut-off for sensitivity analysis. It would be helpful to more explicitly justify this choice in the Methods section e.g., if this is the more commonly used cut-off in the Swedish population, or if you based your choice on the mean GHQ score in your sample. If possible, I would also suggest running a sensitivity analysis using a different cut-off to provide confidence that your results are not being unduly influenced by the use of the low threshold.

We are grateful for this comment. The issue of finding an optimal trade-off between sensitivity and specificity in the determination of the GHQ threshold has been thoroughly investigated and discussed in one of the referenced articles, "Goldberg DP, Oldehinkel T, Ormel J. Why GHQ threshold varies from one place to another. Psychol Med 1998;28:915-21." As suggested therein, we used the mean score to define the threshold; this has now been clarified in the Methods section on page 8.

We used the 0-0-1-1 scoring method (range 0–12) recommended by the creators of the instrument, [22] with poor mental health or ‘GHQ-caseness’ determined by the population mean and defined as a score of 2 or higher. [23]

4) In your Results and Discussion, you should be more cautious in interpreting the stratified IRRs where the confidence intervals have considerable overlap (e.g., for age), and acknowledge that this lack of precision creates difficulties for interpretation of the stratified estimates in your Limitations. This also applies to the IRRs in Tables 4A and 4B, particularly since you do not have confidence intervals around the synergy indices and the confidence intervals again largely overlap (they are especially wide around the estimate for high education/over-indebtedness in men, which should be more explicitly acknowledged when drawing inference from this finding in the Discussion).

We highly value this comment, which made us discover a slight error regarding the confidence intervals in table 4B, which has been corrected. However, the differences between the originally stated confidence intervals are small, the corrected ones are more narrow but still overlapping. Since the analysis was just meant to probe the possibility of a synergistic relation and did not include a formal test regarding this interaction, we have looked over our claims regarding the significance of these findings in the Discussion section on page 20.

Nevertheless, this finding relies on an analysis in which the number of exposed men is small, why the confidence intervals are large. Therefore, this result needs to be replicated in other research.

5) Aside from selection bias, I think the potential for residual confounding is the biggest risk to causal interpretation of your findings, but this is not allocated much discussion in the paper (in fact you consider that instead you may be *over*-adjusting). Based on the literature on income/wealth and health, I suspect there are differences between those exposed and unexposed to over-indebtedness which are not wholly captured by your measured variables of gender, education, country of origin and precarious employment (household composition, individual/household income, area-level deprivation, benefit claimant status, personality/values etc.). More explicit acknowledgement of this as a potential limitation would be welcome.

We have added a sentence regarding residual confounding in the Strengths and Weaknesses section on page 22.

Although we have assessed potential confounding from main socioeconomic background factors known to be linked to over-indebtedness, such as gender, educational level, age, and migration status, we cannot fully rule out residual confounding from other factors.

6) As with the Introduction, there should be more engagement with existing literature in the Discussion, putting your findings into the context of existing research rather than simply restating them. You have some very interesting theories about what your results mean, but I wanted to know

more about how these fit with what is already known or theorised e.g., has your 'life-course' explanation for over-indebtedness patterning by age been reported on elsewhere? What is already known about how over-indebtedness is patterned by social power structures? Are there other literatures on how economic exposures may impact the health of well educated people (particularly men) more severely than those with less education?

We are grateful for these valuable comments. However, the body of research into the health effects of over-indebtedness and poor mental health is to the best of our knowledge very limited. We have once again scrutinized the published literature on this topic, but we did not find more than already included in the references. We have added a sentence concerning this in the Discussion section on page 19.

Previous research provides few examples of how this could be integrated into a coherent theoretical framework regarding health equity. We speculate that...

MINOR COMMENTS

1) On the whole, I think the paper should be proofread a little more finely prior to resubmission as I did notice a few minor errors throughout, for example:

a) In the first line there is a typo – I think economy should be economies?

This has been corrected.

b) In the opening paragraph, the wording should read 'from' a health perspective rather than 'in'

This has been corrected.

c) 'Data' should be plural throughout

This has been corrected.

d) Covid-19 is sometimes capitalised and sometimes not

This has been corrected.

- 2) In the Abstract:
 - a) You should make clear that your sample only includes individuals who are vocationally active, rather than the whole general adult population.

This has been addressed.

- b) You could make clearer the interpretation of the final sentence in Results: in what direction does the synergistic relationship between high levels of education and over-indebtedness impact men?

This has been addressed.

- c) You mention in the Conclusions section that you found precarious employment to be a potential mediator, but you don't refer to this finding in your included Results?

We are grateful for this remark, however we think that this has in fact been addressed by the sentence on page 14 in the Results section.

As a general remark, we find it possible that this is a case of over-adjustment, since all the introduced covariates could be included in the same causal chains, as well as in separate ones, and we have no way to disentangle this.

NB, this version of the sentence has included the modification suggested in comment 4 below.

- 3) You may wish to include the data not shown in tables (e.g. results dichotomised by education level) in an online supplement, if permitted by the journal.

After discussion among the authors, we have chosen not to include this information since we have difficulties fitting it into the existing discussion of the findings.

- 4) In the second-last paragraph of the Results, I would tone down the strength of your statement on over-adjustment after adding the job insecurity variable from being 'very likely' to 'possible' or something else less strong, as I'm not altogether convinced that the bias couldn't theoretically go in either direction

See reply to comment 2c above.

Reviewer: 2

Amity Quinn, University of Calgary

Comments to the Author:

Thank you for the opportunity to review this prospective cohort study on the association of over-indebtedness and mental health. The study is interesting and well-done.

I have a few minor comments:

In the strengths and limitations section, please include which general adult population (e.g., southern Sweden).

This has been corrected.

Methods

- At which assessments (time points) were over-debtedness defined?

We are very grateful for this comment. Over-indebtedness was assessed at all three time points (at baseline in 1999/2000, in 2005, and in 2010) which now has been explicitly clarified in the Methods section. When the variable 'Over-indebtedness at least once in 1999/2000 or 2005' is used in the analyses, this is specified in the table heads.

- Statistical analysis: Could you add some additional details about the interaction analyses?

See reply to Reviewer 1, major comment #4.

Discussion:

- Because information on over-indebtedness is not easily retrievable from registry or administrative health or financial data as you point out, could you expand on what types of policy interventions might be possible to address these problems?

We thank the reviewer for this excellent suggestion. A couple of sentences have been added in the Discussion section on page 21 in order to address this comment.

Because of the growing problem with over-indebtedness, it seems urgent to develop the information systems concerning individual's and household's financial situation, in a manner that high quality information becomes available, not only concerning income but equally concerning financial assets and debt. This, especially since increasing inequality appears to be better mirrored by the latter, rather than the former.

- Could you provide some additional discussion on the generalizability of your findings and the policy context as well as policy implications outside of Sweden?

We are also very grateful for this comment. A sentence has been added in the Discussion section on page 21 in order to address this comment.

A specific policy recommendation is to expand the possibilities to write off debt for the individual, in a similar way that now is possible for business enterprises, in order to prevent a vicious circle of poor mental health and increasing debt, which seems maleficent for all involved stakeholders.

Relatedly, I am curious about safety net policies in Sweden, including inside and outside of the health care system.

We thank the reviewer for this comment. However, we have left this suggestion without action, since we were very unsure what the reviewer actually suggests. A reasonably detailed description of the safety net policies could not in our opinion possibly be squeezed into the manuscript, given the current format, although this might have been of general interest.

Reviewer: 1

Competing interests of Reviewer: None

Reviewer: 2

Competing interests of Reviewer: None

VERSION 2 – REVIEW

REVIEWER	Thomson, Rachel University of Glasgow
REVIEW RETURNED	12-Sep-2022

GENERAL COMMENTS	I feel the authors have adequately addressed my concerns with their revisions to the manuscript, and have no further comments.
--

REVIEWER	Quinn, Amity University of Calgary
REVIEW RETURNED	08-Sep-2022

GENERAL COMMENTS	This revision addressed my previous concerns.
---